# Unravelling the reservoirs for colonisation of infants with *Campylobacter* spp. in rural Ethiopia: protocol for a longitudinal study during a global pandemic and political tensions

Arie H Havelaar ![ORCID],[1] Mussie Brhane ![ORCID],[2] Ibsa Abdusemed Ahmed ![ORCID],[2] Jafer Kedir ![ORCID],[2] Dehao Chen,[1] Loic Deblais,[3] Nigel French ![ORCID],[4] Wondwossen A Gebreyes,[3,5] Jemal Yousuf Hassen,[2] Xiaolong Li,[1] Mark J Manary,[6] Zelealem Mekuria,[3,5] Abdulmuen Mohammed Ibrahim,[2] Bahar Mummed,[2] Amanda Ojeda ![ORCID],[1] Gireesh Rajashekara,[3] Kedir Teji Roba,[2] Cyrus Saleem,[1] Nitya Singh,[1] Ibsa Aliyi Usmane,[2] Yang Yang,[1] Getnet Yimer,[5] Sarah McKune ![ORCID] [1]

For numbered affiliations see end of article.

**Correspondence to**
Professor Arie H Havelaar;
ariehavelaar@ufl.edu

## ABSTRACT

**Introduction** Undernutrition is an underlying cause of mortality in children under five (CU5) years of age. Animal-source foods have been shown to decrease malnutrition in CU5. Livestock are important reservoirs for *Campylobacter* bacteria, which are recognised as risk factors for child malnutrition. Increasing livestock production may be beneficial for improving nutrition of children but these benefits may be negated by increased exposure to *Campylobacter* and research is needed to evaluate the complex pathways of *Campylobacter* exposure and infection applicable to low-income and middle-income countries. We aim to identify reservoirs of infection with *Campylobacter* spp. of infants in rural Eastern Ethiopia and evaluate interactions with child health (environmental enteric dysfunction and stunting) in the context of their sociodemographic environment.

**Methods and analysis** This longitudinal study involves 115 infants who are followed from birth to 12 months of age and are selected randomly from 10 kebeles of Haramaya woreda, East Hararghe zone, Oromia region, Ethiopia. Questionnaire-based information is obtained on demographics, livelihoods, wealth, health, nutrition and women empowerment; animal ownership/management and diseases; and water, sanitation and hygiene. Faecal samples are collected from infants, mothers, siblings and livestock, drinking water and soil. These samples are analysed by a range of phenotypic and genotypic microbiological methods to characterise the genetic structure of the *Campylobacter* population in each of these reservoirs, which will support inference about the main sources of exposure for infants.

**Ethics and dissemination** Ethical approval was obtained from the University of Florida Internal Review Board (IRB201903141), the Haramaya University Institutional Health Research Ethics Committee (COHMS/1010/3796/20) and the Ethiopia National Research Ethics Review Committee (SM/14.1/1059/20). Written informed consent is obtained from all participating

## STRENGTHS AND LIMITATIONS OF THIS STUDY

⇒ Comprehensive characterisation of all *Campylobacter* spp. in a smallholder setting in Africa in humans, livestock and environment.
⇒ First study with a specific focus on transmission of non-thermotolerant *Campylobacter* spp. from livestock reservoirs.
⇒ Longitudinal design allows evaluation of infection and disease dynamics.
⇒ Relatively small sample size, affecting power to detect nutritional impacts of *Campylobacter* colonisation in infants.
⇒ Execution heavily affected by COVID-19 pandemic, civil conflict and other stressors.

households. Research findings will be disseminated to stakeholders through conferences and peer-reviewed journals and through the Feed the Future Innovation Lab for Livestock Systems.

## INTRODUCTION

Undernutrition underlies 45% of worldwide under-five mortality.[1] In Eastern Africa, 5.3% of children under five (CU5) are stunted (a measure of chronic undernutrition, defined as length or height for age Z score< −2) and 34.5% are wasted (a measure of recent undernutrition, defined as length/height for weight Z score< −2).[2] Stunting increases the risks of intermittent illness, reduction of vaccine effectiveness, and suboptimal intellectual development in CU5, and over time is associated with lower income and increased

morbidity and mortality.[3–5] Wasting is a more acute state and is associated with higher mortality risk.[6]

Pathways by which agriculture can alleviate child undernutrition include production of animal-source foods (ASF), household income and women's empowerment.[7] Demand for ASF in Africa is growing rapidly. The livestock master plan (LMP) of Ethiopia aims to increase livestock production for both economic development and the nutritional needs of its growing population.[8 9] In Ethiopia, the LMP is implemented through commercialised and smallholder family production systems.[10] Smallholder systems are often implemented by farmers in low-resource rural settings,[10] where livestock waste management and water, sanitation and hygiene (WaSH) practices are likely to be deficient. As illustrated in the renowned 'F-diagram', the absence of such measures is likely to foster contamination by animal/human faeces, which serve as a reservoir of enteric pathogens.[11]

The infection of enteric pathogens in CU5 is not only associated with symptomatic illness (eg, diarrhoea); importantly, asymptomatic enteric infection may lead to environmental enteric dysfunction (EED).[12] With the realisation that global child undernutrition cannot be solely attributed to deficient diets and diarrhoea, researchers have hypothesised that EED may be the key mediator between environmental exposures to enteric pathogens and undernutrition, calling for expansion of the long-established UNICEF framework of child undernutrition to incorporate poor gut health as an immediate cause of undernutrition.[13–15]

In recent years, two randomised controlled trials found no significant effect of traditional WaSH interventions on stunting.[16] Researchers subsequently called for 'transformative WaSH' to eradicate household faecal contamination. Without tackling environmental contamination (animal waste, soil, poor food hygiene), the negative findings on child growth persisted in a recent intervention implementing this transformative approach, despite finding a partial protective effect against EED.[17] These findings emphasise the necessity of targeting exposure to animal faeces and zoonotic enteric pathogens through effective control measures in future interventions, particularly where livestock farming contributes significantly to rural livelihoods.

A recent risk-benefit analysis of smallholder livestock production summarised a variety of enteric pathogens originating from livestock and associated with EED and undernutrition. Mainly, asymptomatic infections of children with *Campylobacter* spp were found to be associated with EED outcomes of epithelial damage, inflammation and increased permeability of the gut, as well as growth faltering and reduced weight gain.[13] A study in Peru found that asymptomatic infection of *Campylobacter* had a stronger effect size on slowing weight gain than its symptomatic counterpart.[18] In the Etiology, Risk Factors, and Interactions of Enteric Infections and Malnutrition and the Consequences for Child Health (MAL-ED) study, *Campylobacter* spp in child stools detected by Enzyme-Linked

Immuno Assay (ELISA, identifying all species in the genus) had a nearly twofold higher effect on growth faltering than the thermotolerant species *C. jejuni/C. coli* detected by molecular methods, underscoring the potentially important role of non-thermotolerant *Campylobacter* spp. in undernutrition.[12 19] *C. jejuni/C. coli* are the most common causal agents of bacterial enteritis globally,[20] and their animal reservoirs include livestock animals.[21] The epidemiology and reservoirs of non-thermotolerant *Campylobacter* (eg, *C. fetus*, C. *hyointestinalis*, Candidatus *Campylobacter infans*) are less well understood.[22]

We conducted a cross-sectional study of smallholder farming households in Haramaya woreda (district) in rural eastern Ethiopia to measure the prevalence of *Campylobacter* spp., EED and undernutrition in children aged 10–16 months old and related risk factors.[23] A combination of poor child diet, inadequate WaSH conditions and poor livestock waste management potentially accounted for a high prevalence of (predominantly asymptomatic) *Campylobacter* spp colonisation, EED and undernutrition. Although the study was powered for prevalence estimation, current breastfeeding status of the child and child consumption of ASF were both significantly associated with *Campylobacter* spp. colonisation, while access to improved drinking water was protective against EED. Notably, we found a diversity of *Campylobacter* spp. colonised the children's gut, with non-thermotolerant species occurring more frequently and in higher abundance than thermotolerant species.[24] Although previous studies investigated the health impact and reservoirs of non-thermotolerant species, none of them targeted low-income and middle-income country settings and CU5 as study population.[25–27] The findings from our cross-sectional study underscore the need for attribution research examining various livestock species and other potential reservoirs for *Campylobacter* colonisation of infants. This paper presents a longitudinal study protocol and adaptations that have been implemented due to political unrest and the COVID-19 pandemic.

The objectives of the longitudinal study are:

1. To assess the prevalence, species composition, and genomic diversity of thermotolerant and non-thermotolerant *Campylobacter* spp. in infants, adults, livestock and other reservoirs in the Haramaya woreda.
2. To determine the attribution of *Campylobacter* infections in infants to humans, livestock and other reservoirs (ie, drinking water, soil) based on the genetic population structure of *Campylobacter* spp. circulating in these reservoirs.
3. To assess the associations among the presence of *Campylobacter* spp. and the nutritional status (i.e., stunting) of infants in relation to their socioeconomic environment.

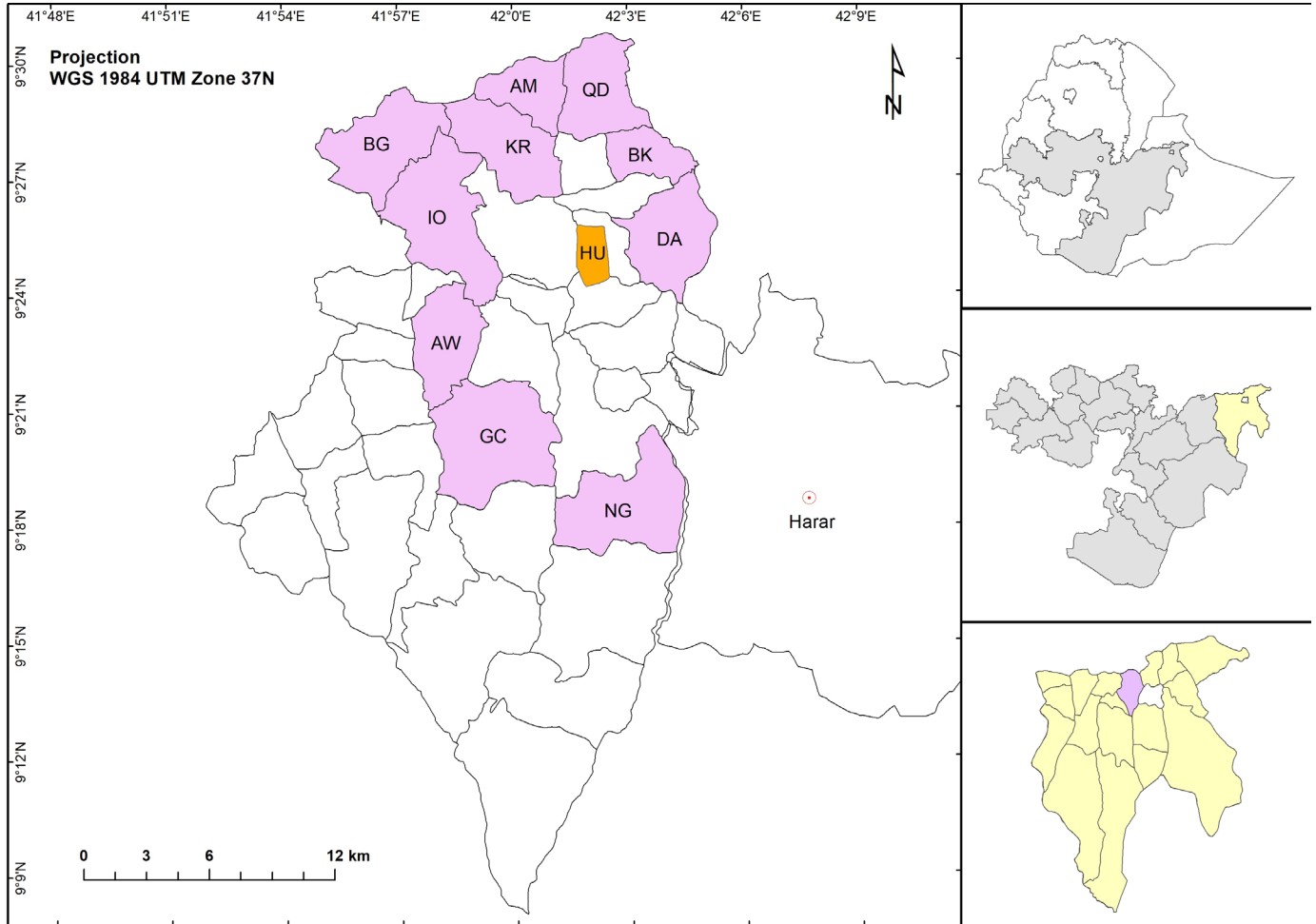

**Figure 1** Geographic location of study area. Right panels from top to bottom: Ethiopia, East Hararghe zone, Oromiya region with Haramaya woreda indicated in purple. Left panel: Haramaya woreda with study sites indicated in purple. Haramaya University (HU) campus is indicated in orange. The urban centre of Harar borders the woreda to the East. AM, Amuma; AW, Adele Walta; BG, Biftu Geda; BK, Bachake; DA, Damota; GC, Gobe Challa; IO, Ifa Oromia; KR, Kuro; NG, Nageya; QD, Qerensa Dereba.

## METHODS AND ANALYSIS

### Study setting

The study is conducted in the rural Haramaya woreda, East Hararghe zone, Oromia Region, Ethiopia (figure 1). Haramaya woreda, at an altitude of 1400–2340 m above sea level, has 36 rural kebeles (the smallest administrative unit in Ethiopia) and three urban kebeles. Khat, vegetables and fruits are important cash crops.

According to the 2016 Ethiopian Demographic and Health Survey, stunting in Haramaya is higher at 45.8% than the national average of 38%.[28] Haramaya University (HU) has established a Health and Demographic Surveillance Site (HDSS) in 12 kebeles in the Haramaya woreda.[29] Overall, 10 of these 12 kebeles (figure 1) will provide the source population for the study. Two kebeles were excluded because of small population size and

**Table 1** Population size and number of participating households in selected kebeles

| Kebele | Population | Participants | Kebele | Population | Participants |
|---|---|---|---|---|---|
| Adele Walta (AW) | 5100 | 11 | Gobe Challa (GC) | 14 300 | 12 |
| Amuma (AM) | 7400 | 12 | Ifa Oromia (IO) | 12 200 | 11 |
| Bachake (BK) | 4900 | 9 | Kuro (KR) | 11 700 | 12 |
| Biftu Geda (BG) | 11 700 | 12 | Nageya (NG) | 11 500 | 12 |
| Damota (DA) | 6600 | 12 | Qerensa Dereba (QD) | 7500 | 12 |

Source: Haramaya University Health and Demographic Surveillance Site (unpublished).

proximity to the urban centre of Harar. The total population of the 10 kebeles is 92 900, with population numbers per kebele ranging between 4900 and 14 300 (table 1).

## Sample size and power

The study is powered for prevalence estimation. A sample of 100 infants or animals allows estimation of a 50% prevalence with precision 10% at 95% confidence, and a power of 80%. We allowed for 20% attrition and aimed to enrol a sample of 120 newborn infants.

Overall, 100 pure cultures of each thermotolerant and non-thermotolerant *Campylobacter* spp. per reservoir (infants, mothers, siblings, four livestock species (cattle, goat, sheep, chicken), water, soil) are analysed by whole-genome sequencing, 1400 in total. This power calculation is based on Smid *et al*,[30] who showed that the precision of attribution is reduced if less than 100 isolates per animal reservoir are used.

## Study population

A birth registry has been developed, which leverages the biannual update of the HDSS by cross-tabulation of expected birthdates in the selected kebeles based on the date of the mother's last menstrual period and the estimated month of pregnancy at the time of the interview. Data on expected deliveries were updated every month, and one DHSS data collector was assigned per kebele to record actual births. Newborn children were randomly selected with the aim to include 12 infants per kebele in the first month after birth. Families were eligible for participation if they had no plans to move out of the Haramaya woreda within 6 months; the mother had resided in the woreda for at least 3 months during pregnancy; and the mother was over 16 years of age when giving birth. Infants were excluded if the birth weight was <2500 g, if the infant or mother required extended stay (more than 4 days) in the hospital after birth, or if the infant had visible congenital abnormality or known serious medical illness or enteropathy, diagnosed by a medical doctor. Enrolment started in December 2020 and was completed by June 2021; 115 infants have successfully been enrolled (table 1). Written informed consent was obtained from all participating households (husband and wife) using a form in the local language (Afan Oromo).

## Infant health measurements and interviews

Infants are followed up from birth to 1 year of age. At enrolment and every 3 months anthropometric measurements (age in days, recumbent length, weight, mid-upper arm circumference (after 6 months of age) and head circumference) are collected.

At the end of the follow-up period (age 12–14 months), EED is detected by a combination of the lactulose absorption test[31] and analysis for faecal myeloperoxidase (MPO) using a commercially available ELISA MPO RUO assay (Alpco, Salem, New Hampshire, USA).

## Collection and management of biological samples

Stool samples are obtained every 4 weeks from all infants and biannually from all mothers, siblings and livestock (one sample from chickens, cattle, goats and sheep per household). Samples from the environment (three soil samples using bootsocks and one sample of drinking water per household) are also collected biannually.

Mothers are provided modified disposable diapers with a clean plastic sheet, sterilised by UV for 10 min and an ice box with ice pack on the day before sample collection. The mother is asked to fit the modified diaper to the infant in the early morning and, after the infant has defecated, to wrap and place the diaper and contents in the ice box. When appropriate, the mother collects a sample of stool from the youngest sibling, as above, and of her own stool in a screw cap bottle. Stool samples are transferred to sterile prelabelled whirl-pack bags, which are transported to the lab in ice boxes.

Samples are transported to a dedicated laboratory at the HU main campus (figure 1) in an ice box within a maximum of 6 hours. Samples for nucleic acid extraction and sequencing are transferred to a nucleic acid stabilising reagent. On arrival in the laboratory, remaining faecal samples are distributed over barcoded tubes and frozen at −80°C, partly with addition of 15% v/v glycerol. Samples for MPO analysis are distributed in barcoded tubes in the field, immediately flash-frozen in liquid nitrogen and stored at −80°C immediately on arrival in the laboratory.

## Data collection and management

All data are collected by trained personnel employed by HU who are proficient in the local language (Afan Oromo), knowledgeable of the local cultural background, and have appropriate scientific backgrounds (health sciences, veterinary sciences and social sciences).

Household questionnaires on demographics; livelihoods; wealth; animal ownership, management and disease; WaSH; infant health and nutrition; and women's empowerment are presented to mothers and fathers two times during the study. Mothers answer a short questionnaire on infant health, vaccinations, breastfeeding practices, antibiotic use and diets during monthly collection of the stool samples. Families who decide to discontinue participation are presented with an exit interview. All data are collected on tablets using the REDCap mobile app and uploaded to a REDCap database, hosted at the University of Florida. The REDCap codebook is available as online supplemental file 1.

## Detection, quantification, isolation and characterisation of *Campylobacter* spp in human stools, livestock faeces and environmental samples

### Molecular detection of *Campylobacter* spp

Genomic DNA is extracted from the field samples using commercial kits. Detection and quantification of *Campylobacter* is performed using genus-specific Taqman real-time PCR[32] and species-specific Sybr Green real-time PCR.[32–36]

Stool samples with a Ct value lower than 35 and 1 Ct value below the negative controls (sterile water, *Salmonella enterica* ssp *enterica* serotype Typhimurium strain LT2 and *Escherichia coli* genomic DNA) are considered positive for *Campylobacter*.

### Detection of *Campylobacter* spp. by culture

Samples positive for *Campylobacter* by qPCR and a random selection of 10% of negative samples are selected for detection of thermophilic and non-thermophilic *Campylobacter* spp. For thermophilic *Campylobacter* spp, decimal dilutions are plated on CHROMagar *Campylobacter* (DRG International, Springfield, New Jersey USA) and incubated for up to 48 hours at 42°C in microaerophilic conditions (85% nitrogen, 10% carbon dioxide, 5% oxygen) using anaerobic jars and GasPak EZ Campy Container System Sachets. For non-thermophilic *Campylobacter* spp, decimal dilutions are plated on Columbia agar supplemented with 5% defibrinated sheep blood, Skirrow supplement (2 μL/mL), amphotericin B (5 μg/mL), cefoperazone (8 μg/mL) and *Campylobacter* growth supplement. The plates are incubated at 37°C for up to 72 hours in anaerobic conditions using anaerobic jars and GasPak anaerobic sachets. If no characteristic growth is observed, samples are enriched in Preston broth (faeces, soil samples for thermophilic *Campylobacter* spp) or Bolton broth (otherwise), incubated and plated as above.

A random selection of up to five presumptively positive colonies from each sample and assay are confirmed by real-time PCR using genus-specific primers.[37] Colonies confirmed as *Campylobacter* spp. are stored in glycerol at −80°C and shipped to the US for speciation using matrix-assisted laser desorption/ionization time-of-flight (MALDI-TOF) mass spectrometry and whole-genome sequence analysis of confirmed isolates.

### Metagenomic sequencing

Metagenomic sequencing will be used to complement detection and genetic characterisation of *Campylobacter* sp in children, mothers, siblings and livestock by with a culture-independent method providing sequences of (genes from) the dominant *Campylobacter* sp. These data will also serve to validate qPCR results and are expected to provide information on infection with other enteric pathogens. We aim to apply 16s rRNA sequencing[38] to infant stool samples after birth and at 4, and 12 months as well as at the time of EED measurement, and to all available stool samples from siblings and mothers. Shotgun sequencing[24] will be applied to a random selection of DNA extracts from human and livestock samples from 45 households.

### Data analysis
#### Objective 1

The prevalence, species composition and genomic diversity of thermotolerant and non-thermotolerant *Campylobacter* spp. in infants, adults, livestock and other reservoirs in the Haramaya woreda will be characterised by descriptive analysis; identification of species and clonal complexes using legacy, core genome and/or whole-genome MLST: phylogenetic analysis,[39] determination of index of diversity, linkage equilibrium and rarefaction analysis.[40] Further, the virulome and resistome of the *Campylobacter* isolates will be analysed using comprehensive virulence factors[41] and antibiotic resistance gene databases.[42] In addition, a functional profiling of the sequenced genome will be predicted.[43] Specificity and sensitivity of qPCR and culture methods for detecting *Campylobacter* at the genus and biological group levels will be evaluated using Bayesian modelling of performance characteristics in the absence of a gold standard.[44–46] Metagenomic sequencing data will be analysed by analysed by latest versions of QIIME V.2,[47] CCMetagen[48] and IDseq pipelines.[49]

We will fit a Bayesian two-state Markov model for the temporal process of colonisation and clearance of *Campylobacter* in each infant. The daily transition probabilities between colonisation and clearance will be regressed on time, demographic variables, household characteristics and most recent *Campylobacter* prevalence among samples from livestock, environment and other household members (sampled biannually).

Spatiotemporal effects will be analysed using boosted regression trees[50] to model the influence of regional environmental variables on the distribution of infections of infants with the *Campylobacter* genus as primary outcome variable. Depending on results, this analysis will also be performed for one or more dominant species. To explore the spatial heterogeneity of effects of covariates on *Campylobacter* infections geographically weighted logistic regression models[51] will be used.

#### Objective 2

The primary outcome variable for this objective is the proportion of *Campylobacter* infections in infants to that is attributed to mothers, siblings, livestock and environmental reservoirs based on the genetic population structure of *Campylobacter* sp circulating in these reservoirs. Source attribution will be based on modified Hald[52] and asymmetric island[53] models using R packages sourceR[54] and islandR (https://github.com/jmarshallnz/islandR). We will also explore the use of whole-genome[55 56] and metagenomic sequencing data[57] for attribution.

#### Objective 3

Data on sociodemographics, livelihoods, economics, environmental health and sanitation will be analysed descriptively, characterising the context within which the study is occurring. Infant health, including breastfeeding practices, infant diet, vaccination status, use of antibiotics and presence of fever and diarrhoea, will also be described. Women's empowerment in agriculture will be assessed using five domains of empowerment generated by the Women's Empowerment in Agriculture Index (WEAI). Within this context, the primary outcome variables (EED biomarkers and nutritional outcome data, including

length-for-age Z score and stunting status) will be examined for their associations with the cumulative burden of *Campylobacter* using generalised linear models and generalised estimating equations, respectively. We will adjust for potential confounders including sex, socioeconomic status and kebele residence, and include potential covariates such as breast feeding, infant health, household WaSH conditions, animal ownership, WEAI and food security. To better estimate the cumulative burden of *Campylobacter* we will extract data from the Bayesian Markov model that simulates the underlying daily presence/absence status of any *Campylobacter* sp in symptomatic or asymptomatic children. We will also perform exploratory analysis of associations between the gut microbiome of infants and infection with *Campylobacter* spp, EED biomarkers and stunting.

## Patient and public involvement

Patients and/or the public were not involved in the design, or conduct, or reporting, or dissemination plans of this research.

## Adaptive project management

The project site was deliberately selected based on the strength of potential partnerships and the realities of local populations; namely, the high rates of malnutrition among rural smallholder farmers, whose livelihoods employ a mixed livestock/crop system. While it was predictable that this type of environment would present significant logistical, technical and sociocultural constraints to research, it was also essential to examine the fundamental research questions. However, from the onset, the project has experienced additional unexpected challenges. The strength of the research partnership and continuous adaptation to the scientific protocol and data collection procedures have allowed the research to advance, despite changes in the local context.

Numerous unforeseen challenges have presented, perhaps most significantly the COVID-19 pandemic. The project start was delayed from April to December 2020. Extensive COVID-19 safety protocols were developed and continuously adapted to allow the research to advance but to prevent COVID-19 transmission among team members and with the study population. Availability and work/transport dynamics of field and laboratory staff were limited by these necessary measures; examples include weekly screening for COVID-19 symptoms of all team members and associated quarantine procedures, and a policy of having only three field team members as passengers (masked) in a vehicle with the masked driver, with windows down. International travel was completely stopped for over a year. Laboratory analyses were also affected by COVID-19 related global supply chain challenges and prolonged customs clearance procedures, resulting in late delivery of critical equipment and supplies. In addition to COVID-19, other unforeseeable challenges for the field work included the escalating conflict in Tigray, which starting in November 2020 and

resulted in a declaration of a national State of Emergency in November 2021, and unusually heavy rains from August to October 2021 resulting in deteriorating road conditions and collapsing bridges. Other more typical setbacks that nonetheless required additional adaptation included reduced availability of the study population during Ramadan (May 2021), personal leave among team members (eg, multiple pregnancies, death of an infant, illness) and concerns about political insecurity surrounding general elections in June 2021.

Consequent to these challenges, the originally planned follow-up period of 18 months was reduced to 12 months. Further adjustments included the timing and distribution of field team members in communities for data collection, number of samples collected per household, number of anthropometric measurements collected per infant. The protocol in this manuscript includes all changes made until the moment of writing (December 2021), with the caveat that because of increased travel times due to checkpoints and roadblocks, planned EED measurements and end-line long interviews are not yet possible and will be attempted later.

Communication, both between the research team and participants, as well as among members of the research team, has been critical to the adaptive management of the project. Interaction with the study population was facilitated by a Community Advisory Board (CAB), established during formative research and made up of representatives of the participating kebeles.[58] The CAB met prior to the start of field work and facilitated the research teams' interactions with local communities. Local CAB members were informed before the team travelled to field sites and were invited into and facilitated critical research-community member communication. This included discussions between the project manager and families who had concerns about ongoing participation in the project.

Realtime information from the CAGED project field teams was instrumental in adapting the project to local norms, expectations and realities, many of which shifted and were reestablished during the COVID-19 pandemic. The field teams shared formal and informal data using a variety of communication platforms. Weekly face-to-face meetings were held for all field teams to discussion and debate approaches to challenges as they occurred; these were followed same day by Zoom-based conversation with Principal Investigators to finalise changes to the weekly plan. The broader team relied heavily on WhatsApp threads for day-to-day communication, field updates or other rapid communication. Zoom-based calls were also used for trainings that targeted the field team, with content ranging from research protocol updates to refresher training on data collection methods, to the importance of COVID-19 vaccines.

Data collection has only proceeded because of the research team's existing partnership, the cultural competency of local team members, and a collective openness to learning and adaptability. In near constant conversation

with researchers on the ground about how stakeholders within the study areas were receiving, reacting and engaging the research programme, the team has effectively adapted the scientific protocol from its original design, without jeopardising its scientific integrity, to answer the research questions. This exercise in adaptive management was facilitated by established trust, transparent communication, local project ownership and a firm commitment to the project's success.

## DISCUSSION

This proposal outlines a detailed research plan for conducting a longitudinal study to understand the complex interactions between livestock and infant health. This will support implementation of the LMP to improve the nutritional status of Ethiopians, while minimising the risk of poor nutritional outcomes associated with EED and impaired linear growth. The knowledge generated in this project will benefit the local population in our study area, who will be provided with knowledge about improving domestic hygiene in relation to management of excreta from livestock and other domestic animals. It also helps the local community and national authorities to understand the determinants of stunting so that they can devise appropriate interventions, which can be evaluated in further studies. Beyond Ethiopia, findings from this study constitute a global public good and will have widespread implications, as the relationship between *Campylobacter* spp, livestock production, EED and infant growth is further explored.

Our study has several limitations. The study has been powered for prevalence estimation and source attribution of *Campylobacter* infections in infants, with a minimum target sample size of 100. This sample size is small to detect associations between nutritional outcomes and putative causal factors such as *Campylobacter* infection. For example, the MAL-ED study enrolled 165–237 infants per study site.[12] Our sample size will allow us to detect relatively strong associations only. However, the MAL-ED study also demonstrated that the effect of the broad group of non-thermotolerant *Campylobacter* sp on linear growth is stronger than the effect of the well-known thermotolerant species *C. jejuni*. Our ability to detect several non-thermotolerant species separately by qPCR rather than at the group level by an all-inclusive immunoassay allows us to zoom in on the likely small number of species whose effect sizes may be larger than that of the whole group. The study is executed in unprecedented times of a major global pandemic and a major civil conflict in Ethiopia, leading to civic unrest, inflation and increased population mobility. This affects the ability to collect all samples and surveys as planned, resulting in an increased level of missing data. We are implementing several approaches to prevent missingness, including active communication with the CAB members, repeat visits and increasing the number of field workers to accomodate the additional workload and driving times. We will use multiple imputation methods to replace the missing data.

Awareness on the association between infant's exposure to zoonotic pathogens in livestock faeces and risk of stunting is generally low, constraining initiatives to decrease exposure to these hazards. We aim to create an enabling policy and institutional environment, ensuring that smallholder livestock production and associated risk of EED receive the attention and investment this problem deserves. For example, farmers are unaware of the danger of livestock excreta or the measures they could take to reduce contamination. Health practitioners are unlikely to be equipped to diagnose and treat *Campylobacter* infections, especially those due to long-term exposure to livestock. Due to lack of awareness, policy-makers and development partners may not prioritise reducing exposure to livestock faeces, in comparison to other food safety, WaSH and broader food security issues. Key strategies for this include the following:

► Mapping of relevant stakeholders operating at different levels (eg, academia, policy-makers, research, extension workers).
► Hosting of stakeholder workshops to discuss on challenges associated with livestock production and associated effects on infant health, and research project proposed solutions and findings to tackle the challenges.
► Feedback of emerging findings from the research project.
► Developing and delivery of knowledge mobilisation strategies that respond to local stakeholders' needs and actively engage them in the uptake and application of research knowledge, using existing government structures to promote safe chicken production and also transfer information and knowledge by training various actors including agricultural extensions, health workers, community health volunteers, etc.

The research findings and lessons from the project will also be documented, synthesised and shared in different ways to inform policy, development practice and to be used as resource material for training farmers, extension workers and future agricultural graduates.

## ETHICS AND DISSEMINATION

Ethical approval was obtained from the University of Florida Internal Review Board (IRB201903141); the Haramaya University Institutional Health Research Ethics Committee (COHMS/1010/3796/20) and the Ethiopia National Research Ethics Review Committee (SM/14.1/1059/20). Written informed consent is obtained from all participating households (husband and wife) using a form in the local language (Afan Oromo). Research findings will be disseminated to community stakeholders, including participants, through the existing CAB. Findings will be disseminated to scientific, academic, policy and development stakeholders through

conferences and peer-reviewed journals and through the Feed the Future Innovation Lab for Livestock Systems.

The Bill and Melinda Gates Foundation, the funder of this trial, requires an open access data policy. Therefore, all manuscripts from this funded work will be open access with the data underlying the published research results available in a public repository. The website https://www.gatesfoundation.org/how-we-work/general-information/open-access-policy provides more information on this policy. The research is part of the Feed the Future Innovation Lab for Livestock Systems. Webinars, research briefs and other targeted dissemination activities will be organised under this umbrella. The project website https://livestocklab.ifas.ufl.edu/projects/caged/ provides access to all results.

A CAB including a representative of the community, religious leaders (imam), woreda and kebele administration, woreda women and children affairs, woreda bureau of health and agriculture, kebele health, and agricultural extension workers has been established to guide the research team for better understanding of local context and entry to the community and is regularly engaged in the research. Only the project manager at Haramaya University and the data manager at the University of Florida will have access to personally identifiable information in the REDCap database. Any data shared among researchers within the project will be deidentified and blinded. Materials and Data Transfer Agreements assure confidentiality of data when exchanged with international partners and others.

We follow the current standard of care in Ethiopia for the management of sick children. Accordingly, treatment will not be provided for asymptomatic *Campylobacter* infections. All infants with acute disease (diarrhoea, fever) or severe acute malnutrition are linked with the nearby health facility. Transportation to the facility and referral arrangements are offered by the project.

**Author affiliations**
[1]University of Florida, Gainesville, Florida, USA
[2]Haramaya University, Dire Dawa, Ethiopia
[3]The Ohio State University, Columbus, Ohio, USA
[4]Massey University, Palmerston North, New Zealand
[5]Ohio State Global One Health LLC, Addis Ababa, Ethiopia
[6]Washington University in St Louis, St Louis, Missouri, USA

**Acknowledgements** We thank Yenenesh Demisse, Saeda Mukta Mohammed, Efrah Ali Yusuf, Abadir Jemal Seran, Kuniuza Adem Umer, Mawerdi Mohammed Dawit Kedir Abdi Hassen and Belisa Usmael Ahmedo (Haramaya University) for their contributions to the project.

**Contributors** All authors critically reviewed and approved the final version of this manuscript and had final responsibility for the decision to submit for publication. Conceptualisation: AHH, NF, WAG, JYH, MJM, GR, SM. Data curation: DC, XL, BM, AO, CS, NS. Formal analysis: DC, LD, XL, ZM, KTR, NS, YY, SM. Funding acquisition: AHH, NF, WAG, JYH, MJM, GR, SM. Investigation: MBA, IAA, JKA, LD, BM, AO, IAU. Methodology: AHH, LD, MJM, ZM, GR, NS, YY. Administration: AHH, NF, WAG, JYH, MJM, AMI, GR, GY, SM. Resources: AHH, NF, WAG, JYH, MJM, AMI, GR, SM. Supervision: AHH, NF, WAG, JYH, MJM, AMI, GR, KTR, GY, SM. Writing—original draft: AHH, DC, SM. Writing—review & editing: MBA, IAA, JKA, LD, NF, XL, MJM, ZM, AMI, BM, AO, GR, KTR, CS, NS, IAU, YY, GY. AHH is acting as guarantor.

**Funding** This project is funded by the United States Agency for International Development Bureau for Food Security under Agreement #AID-OAA-L-15-00003 as part of Feed the Future Innovation Lab for Livestock Systems, and by the Bill & Melinda Gates Foundation OPP#1175487. Under the grant conditions of the Foundation, a Creative Commons Attribution 4.0 Generic License has already been assigned to the Author Accepted Manuscript version that might arise from this submission. Any opinions, findings, conclusions or recommendations expressed here are those of the authors alone. Research reported in this publication was supported by the University of Florida Clinical and Translational Science Institute, which was supported in part by the NIH National Center for Advancing Translational Sciences under award number UL1TR001427.

**Map disclaimer** The inclusion of any map (including the depiction of any boundaries therein), or of any geographic or locational reference, does not imply the expression of any opinion whatsoever on the part of *BMJ* concerning the legal status of any country, territory, jurisdiction or area or of its authorities. Any such expression remains solely that of the relevant source and is not endorsed by *BMJ*. Maps are provided without any warranty of any kind, either express or implied.

**Competing interests** None declared.

**Patient and public involvement** Patients and/or the public were not involved in the design, or conduct, or reporting, or dissemination plans of this research.

**Patient consent for publication** Not applicable.

**Provenance and peer review** Not commissioned; externally peer reviewed.

**ORCID iDs**
Arie H Havelaar http://orcid.org/0000-0002-6456-5460
Mussie Brhane http://orcid.org/0000-0001-9552-2351
Ibsa Abdusemed Ahmed http://orcid.org/0000-0002-6529-587X
Jafer Kedir http://orcid.org/0000-0001-7596-9324
Nigel French http://orcid.org/0000-0002-6334-0657
Amanda Ojeda http://orcid.org/0000-0002-5059-0008
Sarah McKune http://orcid.org/0000-0002-3646-5921

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
