## [Reviewer comments · BMJ Open]

ARTICLE DETAILS

TITLE (PROVISIONAL)	Unraveling the reservoirs for colonization of infants with Campylobacter spp. in rural Ethiopia: protocol for a longitudinal study during a global pandemic and political tensions
AUTHORS	Havelaar, AH; Adhanom, Mussie; Ahmed, Ibsa Abdusemed; Amin, Jafer; Chen, Dehao; Deblais, Loic; French, Nigel; Gebreyes, Wondwossen A.; Hassen, Jemal Yousuf; Li, Xiaolong; Manary, Mark J; Mekuria, Zelealem; Ibrahim, Abdulmuen; Mammed, Bahar; Ojeda, Amanda; Rajashekara, Gireesh; Roba, Kedir; Saleem, Cyrus; Singh, Nitya; Usmane, Ibsa Aliyi; Yang, Yang; Yimer, Getnet; McKune, Sarah

VERSION 1 – REVIEW

REVIEWER	S. M. Kabir Osaka Prefecture University
REVIEW RETURNED	22-Feb-2022

GENERAL COMMENTS	- Page 9, Line 4: Campylobacter spp. will be Campylobacter spp. - Page 9, Line 26: 5 % defibrinated sheep blood will be 5% defibrinated sheep blood
--

REVIEWER	Bineyam Taye Colgate University Division of Natural Sciences and Mathematics, Biology
REVIEW RETURNED	13-May-2022

GENERAL COMMENTS	Abstract Background:” Increasing livestock production may be beneficial for improving the nutrition of children, but these benefits may be negated by increased exposure to Campylobacter.” I don’t think this is a logical claim. I believe the background should focus on the link between Campylobacter /or infection and malnutrition. Objectives Objective 3 looks very vague. What do you want to measure as outcomes? Assessment of demographic risk factors associated with Campylobacter could be easily answered as part of the objective 1 Although the authors tried to justify their study from a malnutrition perspective, there is no clear objective stating how and what type of nutrition indicators will be targeted (Stunting, underweight, wasting) Methods Child health measures Clearly show how you are going to monitor/follow the study subjects,
---

	like the frequency and interval. Mention the name of the specific laboratory where samples will be transported immediately after collection. Sample size and power You mentioned enrolling 120 children but failed to indicate the number and type of animals. Data analysis Add a data analysis plan for your nutritional outcomes. I would suggest that you list the primary outcome variables under a new subheading before the data analysis. While you, the project claimed to look at the interaction of campylobacter and livestock animals but ignored the role of poultry as a source of Campylobacter. Do you have ways to include poultry-related variables?
--	---

REVIEWER	David Campbell Sheffield Children's NHS Foundation Trust, Paediatric Gastroenterology
REVIEW RETURNED	07-Jun-2022

GENERAL COMMENTS	Thank you for submitting this protocol, it will certainly be helpful and a useful contribution to understanding. I would like to suggest, that the association between enteric health and campylobacter species may be an indirect or direct one. The measurement of campylobacter spp is one of many other potentially important microbiological factors that may be detected on a full analysis of the microbiome. That does not undermine the importance of what you are doing, but framing the results in a manner that helps the reader understand there are potentially further interactions to consider (which I am sure you already have in mind), would be helpful. It was a pleasure to read your manuscript and thank you for submitting it.
--

REVIEWER	Towhid Hasan School of Science, Monash University Malaysia
REVIEW RETURNED	29-Jun-2022

GENERAL COMMENTS	Thanks for considering me to review this protocol. The current proposal has merit in conducting a longitudinal study to comprehend the association between livestock and infant health and can be applied in other regions of the world. I recommend publication of this protocol; however, there is an issue that needs to be considered. The title mentions "infants," but in many places of the proposal, "children" has been used. For example, the word "young children" has been used in many places, which might create confusion (since young children can be thought of as any children under five years of age). As this study is concerned with children aged between birth to 12 months, I suggest being consistent in using terms.
--

VERSION 1 – AUTHOR RESPONSE

Reviewer: 1

Dr. S. M. Kabir, Osaka Prefecture University

Comments to the Author:

- Page 9, Line 4: *Campylobacter* spp. will be *Campylobacter* spp.

Author response: We use *Campylobacter* (in italics) often without the spp. addition, which is common in microbiology literature. We only add spp. if we describe the taxonomic position of the genus.

- Page 9, Line 26: 5 % defibrinated sheep blood will be 5% defibrinated sheep blood

Author response: Corrected, thank you for spotting the typo.

Reviewer: 2

Dr. Bineyam Taye, Colgate University Division of Natural Sciences and Mathematics

Comments to the Author:

Abstract

Background:” Increasing livestock production may be beneficial for improving the nutrition of children, but these benefits may be negated by increased exposure to *Campylobacter*.”

I don't think this is a logical claim. I believe the background should focus on the link between *Campylobacter* /or infection and malnutrition.

Author response:

The causal web between livestock production and possible negative effects on child growth as mediated by zoonotic pathogens in livestock feces is complex and multifactorial. Our research is aimed at improving understanding of this web. We have described our current understanding in reference 13 and summarized the key aspects in the Introduction. The Abstract offers only a snapshot of these arguments, and we understand that it may be difficult to envision the full problem if summarized in just a few sentences. Nevertheless, as the link between livestock and child health is the main driver of our work, we respectfully suggest accepting this part of the abstract as is.

Objectives

Objective 3 looks very vague. What do you want to measure as outcomes? Assessment of demographic risk factors associated with *Campylobacter* could be easily answered as part of the objective 1

Author response: Indeed, the factors associated with *Campylobacter* colonization will be addressed under objective 1. We have replaced “health status” by “nutritional status (i.e., stunting)”, which is our primary indicator for health impacts.

Although the authors tried to justify their study from a malnutrition perspective, there is no clear objective stating how and what type of nutrition indicators will be targeted (Stunting, underweight, wasting)

Author response: Please see previous response.

Methods

Child health measures

Clearly show how you are going to monitor/follow the study subjects, like the frequency and interval.

Author response:

We have added “Infants are followed up from birth to one year of age”; frequencies of follow-up were already specified in the original text.

Mention the name of the specific laboratory where samples will be transported immediately after collection.

Author response: We have added "a dedicated laboratory at the Haramaya University main campus (Figure 1)".

Sample size and power

You mentioned enrolling 120 children but failed to indicate the number and type of animals.

Author response:

We have added more details; the full paragraph now reads "Stool samples are obtained every four weeks from all infants and biannually from all mothers, siblings and livestock (one sample from chickens, cattle, goats and sheep per household). Samples from the environment (three soil samples using bootsocks and one sample of drinking water per household) are also collected biannually."

Data analysis

Add a data analysis plan for your nutritional outcomes.

Author response: We have included more details on the nutritional data analysis as requested.

I would suggest that you list the primary outcome variables under a new subheading before the data analysis.

Author response:

Primary outcome indicators have been identified in each section on data analysis.

While you, the project claimed to look at the interaction of campylobacter and livestock animals but ignored the role of poultry as a source of Campylobacter. Do you have ways to include poultry-related variables?

Author response:

The only poultry species in the study area is chickens, and these are included in our sampling plan. In addition, the long survey includes detailed questions on livestock ownership and management, including poultry. For readers interested in details of our methodology, we have added the long survey questionnaire as supplementary information.

Reviewer: 3

Dr. David Campbell, Sheffield Children's NHS Foundation Trust

Comments to the Author:

Thank you for submitting this protocol, it will certainly be helpful and a useful contribution to understanding. I would like to suggest, that the association between enteric health and campylobacter species may be an indirect or direct one. The measurement of campylobacter spp is one of many other potentially important microbiological factors that may be detected on a full analysis of the microbiome. That does not undermine the importance of what you are doing, but framing the results in a manner that helps the reader understand there are potentially further interactions to consider (which I am sure you already have in mind), would be helpful.

It was a pleasure to read your manuscript and thank you for submitting it.

Author response:

Thank you for your encouraging comments. We are collecting detailed socio-economic data to better understand the complex web of factors that affect exposure to Campylobacter and their health

impacts. As mentioned in our response to the editor, we have received additional funding to support metagenomic sequencing, which will allow us to also include the microbiome in our analyses.

Reviewer: 4

Mr. Towhid Hasan, School of Science, Monash University Malaysia, Department of Food Technology and Nutrition Science, Noakhali Science and Technology University

Comments to the Author:

Thanks for considering me to review this protocol. The current proposal has merit in conducting a longitudinal study to comprehend the association between livestock and infant health and can be applied in other regions of the world. I recommend publication of this protocol; however, there is an issue that needs to be considered. The title mentions “infants,” but in many places of the proposal, “children” has been used. For example, the word “young children” has been used in many places, which might create confusion (since young children can be thought of as any children under five years of age). As this study is concerned with children aged between birth to 12 months, I suggest being consistent in using terms.

Author response:

Thank you for noting this inconsistency, we have replaced child by infant in the manuscript where appropriate. As references cited in the Introduction generally describe children under five years of age, we have not made these edits there.